# Improving Certified Robustness via Statistical Learning with Logical Reasoning

**Zhuolin Yang**[*]
UIUC
zhuolin5@illinois.edu

**Zhikuan Zhao**[*]
ETH Zürich
zhikuan.zhao@inf.ethz.ch

**Boxin Wang**
UIUC
boxinw2@illinois.edu

**Jiawei Zhang**
UIUC
jiaweiz7@illinois.edu

**Linyi Li**
UIUC
linyi2@illinois.edu

**Hengzhi Pei**
UIUC
hpei4@illinois.edu

**Bojan Karlaš**
ETH Zürich
karlasb@inf.ethz.ch

**Ji Liu**
Kwai Inc.
ji.liu.uwisc@gmail.com

**Heng Guo**
University of Edinburgh
hguo@inf.ed.ac.uk

**Ce Zhang**
ETH Zürich
ce.zhang@inf.ethz.ch

**Bo Li**
UIUC
lbo@illinois.edu

## Abstract

Intensive algorithmic efforts have been made to enable the rapid improvements of certificated robustness for complex ML models recently. However, current robustness certification methods are only able to certify under a limited perturbation radius. Given that existing *pure data-driven* statistical approaches have reached a bottleneck, in this paper, we propose to integrate statistical ML models with knowledge (expressed as logical rules) as a *reasoning* component using Markov logic networks (MLN), so as to further improve the overall certified robustness. This opens new research questions about certifying the robustness of such a paradigm, especially the reasoning component (e.g., MLN). As the first step towards understanding these questions, we first prove that the computational complexity of certifying the robustness of MLN is #P-hard. Guided by this hardness result, we then derive the first certified robustness bound for MLN by carefully analyzing different model regimes. Finally, we conduct extensive experiments on five datasets including both high-dimensional images and natural language texts, and we show that the certified robustness with knowledge-based logical reasoning indeed significantly outperforms that of the state-of-the-arts.

## 1 Introduction

Given extensive studies on adversarial attacks against ML models recently [3, 13, 39, 24, 64, 23, 53], building models that are robust against such attacks is an important and emerging topic. Thus, a plethora of *empirical defenses* have been proposed to improve the ML robustness [30, 59, 22, 43, 53, 52]; however, most of these are attacked again by stronger adaptive attacks [3, 1, 46]. To end such repeated security cat-and-mouse games, there is a line of research focusing on developing *certified defenses* for DNNs under certain adversarial constraints [8, 26, 25, 54, 28, 61, 27, 60, 58].

---

[*]The first two authors contribute equally to this work.

36th Conference on Neural Information Processing Systems (NeurIPS 2022).

Though promising, existing *certified defenses* are restricted to certifying the model robustness within a limited $\ell_p$ norm bounded perturbation radius [56, 8]. One potential reason for such limitations for existing robust learning approaches is inherent in the fact that most of them have been treating machine learning as a "pure data-driven" technique that solely depends on a given training set, without interacting with the rich exogenous information such as domain knowledge (e.g., *a stop sign should be of the octagon shape*); while we know human, who has knowledge and inference abilities, is resilient to such attacks. Indeed, a recent seminal work [17] illustrates that integrating knowledge rules can significantly improve the *empirical* robustness of ML models, while leaving the *certified robustness* completely unexplored.

In this paper, we follow this promising `Learning+Reasoning` paradigm [17] and conduct, to our best knowledge, the first study on certified robustness for it. Actually, such a `Learning+Reasoning` paradigm has enabled a diverse range of applications [38, 62, 2, 37, 32, 55, 17] including the ECCV'14 best paper [10] that encodes label relationships as a probabilistic graphical model and improves the *empirical* performance of deep neural networks on ImageNet. In this work, we first provide a concrete *Sensing-reasoning pipeline* following such paradigm to integrate statistical learning with logical reasoning as illustrated in Figure 1. In particular, the *Sensing Component* contains a set of statistical ML models such as deep neural networks (DNNs) that output their predictions as a set of Boolean random variables; and the *Reasoning Component* takes this set of Boolean random variables as inputs for logical inference models such as Markov logic networks (MLN) [40] or Bayesian networks (BN) [36] to produce the final output. We then prove the hardness of certifying the robustness of such a pipeline with MLN for reasoning. Finally, we provide an algorithm to certify the robustness of sensing-reasoning pipeline and we evaluate it on five datasets including both image and text data.

However, certifying the robustness of sensing-reasoning pipeline is challenging, especially given the inference complexity of the reasoning component. Our goal is to take the first step in tackling this challenge. In particular, the robustness certification of sensing-reasoning pipeline can be expressed as the confidence interval of the marginal probability for the final output of *reasoning component*. That is to say, we can use existing state-of-the-art methods to certify the robustness of the sensing component that contains DNNs or ensembles [8, 42, 57]. Thus, to provide the end-to-end certification for the whole pipeline, what is left is to understand *how to certify the reasoning component*, which is the focus of this work.

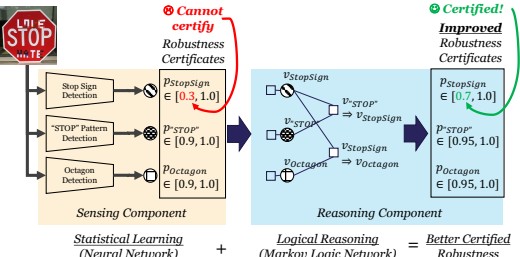

Figure 1: The sensing-reasoning pipeline, i.e., a *sensing component* consists of DNNs and a *reasoning component* is constructed as MLN. The goal of this paper is to provide certified robustness for such a pipeline, especially the reasoning component.

Compared with previous efforts focusing on certified robustness of neural networks, the reasoning component brings its own challenges and opportunities. Different from a neural network whose inference can be executed in polynomial time, many reasoning models such as MLN can be #P-complete for inference. However, as many reasoning models define a probability distribution in the exponential family, we have more functional structures that could potentially make the robustness optimization (which essentially solves a min-max problem) easier. *In this paper, we provide the first treatment to this problem characterized by these unique challenges and opportunities.*

We focus on MLN as the *reasoning component*, and explored three technical questions, each of which corresponds to a technical contribution of this work.

*1. Is certifying robustness for the reasoning component feasible when the inference of the reasoning component is #P-hard?* (Section 3) Before any concrete algorithm can be proposed, it is important to understand the computational complexity of the robustness certification. We first prove that the famous problem of counting in statistical inference [49] can be reduced to the problem of checking the certified robustness of general reasoning components and MLN. Therefore, checking certified robustness is no easier than counting on the same family of distribution. In other words, when the reasoning component is a graphical model such as MLN, checking certified robustness is no easier than calculating the partition function of the underlying graphical model, which is #P-hard.

*2. Can we efficiently reason about the certified robustness for the reasoning component when given an oracle for statistical inference?* (Section 4.2) Given the above hardness result, we focus on certifying

the robustness given an inference oracle. However, even when statistical inference can be done by a given oracle [21, 18], it is still challenging to certify the robustness of MLN. Our second technical contribution is to develop such an algorithm for MLN as the reasoning component. We prove that providing certified robustness for MLN is possible because of the structure inherent in the probabilistic graphical models and distributions in the exponential family, which could lead to monotonicity and convexity properties under certain conditions for solving the certification optimization.

*3. Can a reasoning component improve the certified robustness compared with the state-of-the-art certification methods?* (Section 5) We test our algorithms on multiple sensing-reasoning pipelines, in which the sensing components contain the state-of-the-art *deep neural networks*. We construct these pipelines to cover a range of applications including image classification and natural language processing tasks. We show that based on our certification method on the reasoning component, the knowledge-enriched sensing-reasoning pipelines achieves significantly higher certified robustness than the state-of-the-art certification methods for DNNs.

The rest of the paper is organized as follows. We will first introduce the design of the sensing-reasoning pipeline in Section 2.1, followed by concrete illustrations taking the Markov Logic Networks as an example of the reasoning component in Section 2.2. Next, to certify the robustness of the sensing-reasoning pipeline, especially for the reasoning component, we first prove that certifying the robustness of the reasoning component itself is #P-complete (Section 3), and therefore we propose a certification algorithm to upper/lower bound the certification in Section 4, We provide the evaluation of our robustness certification considering different tasks in Section 5.

## 2 Robust Statistical Learning with Logical Reasoning

In this section, we first provide a sensing-reasoning pipeline and then formally defined its certified robustness, and particularly links it to certifying the robustness for the reasoning component.

### 2.1 Sensing-Reasoning Pipeline

A sensing-reasoning pipeline contains a set of $n$ sensors $\{S_i\}_{i \in [n]}$ and a reasoning component $R$. Each sensor is a binary classifier (for multi-class classifier it corresponds to a group of sensors) — given an input data example $X$, each of the sensor $S_i$ outputs a probability $p_i(X)$ (i.e., if $S_i$ is a neural network, $p_i(X)$ represents its output after the final softmax layer). The reasoning component takes the outputs of all sensing models as its inputs, and outputs a new Boolean random variable $R(\{p_i(X)\}_{i \in [n]})$.

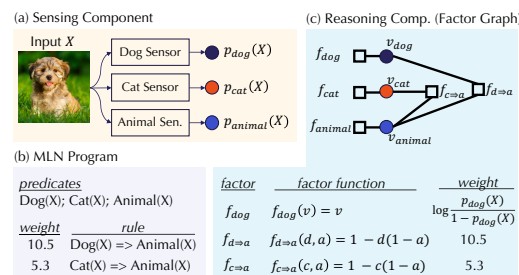

Figure 2: A sensing-reasoning pipeline with MLN as the reasoning component.

One natural choice of the reasoning component is to use a probabilistic graphical model (PGM). In the following subsection, we will make the reasoning component $R$ more concrete by instantiating it as a Markov logic network (MLN). The output of a sensing-reasoning pipeline on the input data example $X$ is the expectation of the output of reasoning component $R$: $\mathbb{E}[R(\{p_i(X)\}_{i \in [n]})]$.

**Example.** A sensing-reasoning pipeline provides a generic, principled way of integrating domain knowledge with the output of statistical predictive models such as neural networks. One such example is [10] the task of ImageNet classification. Here each sensing model corresponds to the classifier for one specific class in ImageNet, e.g., $S_{dog}(X)$ and $S_{animal}(X)$. The reasoning component then encodes domain knowledge such that "*If an image is classified as a dog then it must also be classified as an animal*" using a PGM. There is no prior work considering the certified robustness of such a knowledge-enabled ML pipeline. Figure 2 illustrates a concrete sensing-reasoning pipeline, in which the reasoning component is implemented as an MLN.

### 2.2 Reasoning Component as Markov Logic Networks

Given the generic definition of a sensing-reasoning pipeline, one can use different models to implement the reasoning components. In this paper, we focus on Markov logic networks (MLN), which is a popular way to define a probabilistic graphical model using first-order logic [41]. Concretely, we define the reasoning component implemented as an MLN, which contains a set of weighted first-order logic rules, as illustrated in Figure 2(b). After grounding, an MLN defines a joint probabilistic distribution among a collection of random variables, as illustrated in Figure 2(c). We adapt the

standard MLN semantics to a sensing-reasoning pipeline and use a slightly more general variant compared with the original MLN [41]. Each MLN program corresponds to a factor graph — Due to the space limitation, we will not discuss the grounding part and point the readers to [41]. We focus on defining the result after grounding, i.e., the factor graph.

Specifically, a grounded MLN is a factor graph $\mathcal{G} = (\mathcal{V}, \mathcal{F})$, where $\mathcal{V}$ is a set of Boolean random variables. Specific to a sensing-reasoning pipeline, there are two types of random variables $\mathcal{V} = \mathcal{X} \cup \mathcal{Y}$:

1. **Interface Variables** $\mathcal{X} = \{x_i\}_{i \in [n]}$**:** Each sensing model $S_i$ corresponds to one interface variable $x_i$ in the grounded factor graph;

2. **Interior variables** $\mathcal{Y} = \{y_i\}_{i \in [m]}$ are other variables introduced by the MLN model.

Each factor $F \in \mathcal{F}$ contains a weight $w_F$ and a factor function $f_F$ defined over a subset of variables $\bar{\mathbf{v}}_F \subseteq \mathcal{V}$ that *returns* $\{0, 1\}$. There are two sets of factors $\mathcal{F} = \mathcal{G} \cup \mathcal{H}$:

1. **Interface Factors** $\mathcal{G}$**:** For each interface variable $x_i$, we create one interface factor $G_i$ with weight $w_{G_i} = \log[p_i(X)/(1 - p_i(X))]$ and factor function $f_{G_i}(a) = \mathcal{I}[a = 1]$ defined over $\bar{\mathbf{v}}_{f_{G_i}} = \{x_i\}$.

2. **Interior Factors** $\mathcal{H}$ are other factors introduced by the MLN program.

*Remarks: MLN-specific Structure.* Our result applies to a more general family of factor graphs and are not necessarily specific to those grounded by MLN. Moreover, MLN provides an intuitive way of grounding such a factor graph with domain knowledge, and factor graphs grounded by MLN have certain properties that we will use later, e.g., all factors only return non-negative values, and there are no unusual weight sharing structures.

The above factor graph defines a joint probability distribution among all variables $\mathcal{V}$. We define a *possible world* as a function $\sigma : \mathcal{V} \mapsto \{0, 1\}$ that corresponds to one possible assignment of values to each random variable. Let $\Sigma$ denote the set of all (exponentially many) possible worlds.

The *statistical inference* process of a reasoning component implemented using MLNs [41] computes the marginal probability of a given variable $v \in \mathcal{V}$:

$$\mathbb{E}[R_{MLN}(\{p_i(X)\}_{i \in [n]})] = \mathbf{Pr}\,[v = 1] = Z_1(\{p_i(X)\}_{i \in [n]})/Z_2(\{p_i(X)\}_{i \in [n]})$$

where the partition functions $Z_1$ and $Z_2$ are defined as

$$Z_1(\{p_i(X)\}_{i \in [n]}) = \sum_{\sigma \in \Sigma \wedge \sigma(v) = 1} \exp\left\{\sum_{G_i \in \mathcal{G}} w_{G_i}\sigma(x_i) + \sum_{H \in \mathcal{H}} w_H f_H(\sigma(\bar{\mathbf{v}}_H))\right\}$$

$$Z_2(\{p_i(X)\}_{i \in [n]}) = \sum_{\sigma \in \Sigma} \exp\left\{\sum_{G_i \in \mathcal{G}} w_{G_i}\sigma(x_i) + \sum_{H \in \mathcal{H}} w_H f_H(\sigma(\bar{\mathbf{v}}_H))\right\}$$

**Why** $w_{G_i} = \log[p_i(X)/(1 - p_i(X))]$**?** When the MLN does not introduce any interior variables and interior factors, it is easy to see that setting $w_{G_i} = \log[p_i(X)/(1 - p_i(X))]$ ensures that the marginal probability of each interface variable equals to the output of the original sensing model $p_i(X)$. This means that if we do not have additional knowledge in the reasoning component, the pipeline outputs the *same* distribution as the original sensing component.

**Learning Weights for Interior Factors?** In this paper, we view all weights for interior factors as hyperparameters. These weights can be learned by maximizing the likelihood with weight learning algorithms for MLNs [29].

**Beyond Marginal Probability for a Single Variable.** We have assumed that the output of a sensing-reasoning pipeline is the marginal probability distribution of a given random variable in the grounded factor graph. However, our result can be more general — given a function over possible worlds and outputs $\{0, 1\}$, the output of a pipeline can be the marginal probability of such a function. This will not change the algorithm that we propose later.

## 3 Hardness of Certifying Reasoning Robustness

*Given a reasoning component R, how hard is it to reason about its robustness?* In this section, we aim at understanding this fundamental question. In order to provide the certified robustness of the reasoning component, which is defined as the lower bound of model predictions for inputs considering

an adversarial perturbation with bounded magnitude [8], we need to analyze the hardness of this certification problem first. Specifically, we present the hardness results of determining the robustness of the reasoning component defined above, before we can provide our certification algorithm in Section 4.2. We start by defining the counting [49] and robustness problems on general distribution. We prove that counting can be reduced to checking for reasoning robustness, and hence the latter is at least as hard; We then prove the complexities of reasoning with MLN.

## 3.1 Harness of Certifying General Reasoning Model

Let $X = \{x_1, x_2, \ldots, x_n\}$ be a set of variables. Let $\pi_\alpha$ be a distribution over $D^{[n]}$ defined by a set of parameters $\alpha \in P^{[m]}$, where $D$ is the domain of variables, either discrete or continuous, and $P$ is the domain of parameters. We call $\pi$ *accessible* if for any $\sigma \in D^{[n]}$, $\pi_\alpha(\sigma) \propto w(\sigma; \alpha)$, where $w : D^{[n]} \times P^{[m]} \to \mathbb{R}_{\geq 0}$ is a polynomial-time computable function. We will restrict our attention to accessible distributions only. We use $Q : D^{[n]} \to \{0, 1\}$ to denote a Boolean query, which is a polynomial-time computable function. We define the following two oracles:

**Definition 1** (COUNTING). Given input polynomial-time computable weight function $w(\cdot)$ and query function $Q(\cdot)$, parameters $\alpha$, a real number $\epsilon > 0$, a COUNTING oracle outputs a real number $Z$ that

$$1 - \epsilon \leq \frac{Z}{\mathbb{E}[\sigma \sim \pi_\alpha]Q(\sigma)} \leq 1 + \epsilon.$$

**Definition 2** (ROBUSTNESS). Given input polynomial-time computable weight function $w(\cdot)$ and query function $Q(\cdot)$, parameters $\alpha$, two real numbers $\epsilon > 0$ and $\delta > 0$, a ROBUSTNESS oracle decides, for any $\alpha' \in P^{[m]}$ such that $\|\alpha - \alpha'\|_\infty \leq \epsilon$, whether the following is true:

$$|\mathbb{E}[\sigma \sim \pi_\alpha]Q(\sigma) - \mathbb{E}[\sigma \sim \pi_{\alpha'}]Q(\sigma)| < \delta.$$

We can prove that ROBUSTNESS is at least as hard as COUNTING by a reduction argument.

**Theorem 1** (COUNTING $\leq_t$ ROBUSTNESS). *Given polynomial-time computable weight function $w(\cdot)$ and query function $Q(\cdot)$, parameters $\alpha$ and real number $\epsilon > 0$, the instance of* COUNTING, *$(w, Q, \alpha, \epsilon)$ can be determined by up to $O(1/\varepsilon_c^2)$ queries of the* ROBUSTNESS *oracle with input perturbation $\epsilon = O(\varepsilon_c)$.*

*Proof-sketch.* We define the partition function $Z_i := \sum_{\sigma:Q(\sigma)=i} w(\sigma; \alpha)$ and $\mathbb{E}[\sigma \sim \pi_\alpha]Q(\sigma) = Z_1/(Z_0 + Z_1)$. We then construct a new weight function $t(\sigma; \alpha) := w(\sigma; \alpha) \exp(\beta Q(\sigma))$ by introducing an additional parameter $\beta$, such that $\tau_\beta(\sigma) \propto t(\sigma; \beta)$, and $\mathbb{E}[\sigma \sim \tau_\beta]Q(\sigma) = \frac{e^\beta Z_1}{Z_0 + e^\beta Z_1}$. Then we consider the perturbation $\beta' = \beta \pm \epsilon$, with $\epsilon = O(\varepsilon_c)$ and query the ROBUSTNESS oracle with input $(t, Q, \beta, \epsilon, \delta)$ multiple times to perform a binary search in $\delta$ to estimate $|\mathbb{E}[\sigma \sim \pi_\beta]Q(\sigma) - \mathbb{E}[\sigma \sim \pi_{\beta'}]Q(\sigma)|$. Perform a further "outer" binary search to find the $\beta$ which maximizes the perturbation. This yields a good estimator for $\log \frac{Z_0}{Z_1}$ which in turn gives $\mathbb{E}[\sigma \sim \pi_\alpha]Q(\sigma)$ with $\varepsilon_c$ multiplicative error. We leave detailed proof to Appendix A.

## 3.2 Hardness of Certifying Markov Logic Networks

Given Theorem 1, we can now state the following result specifically for MLNs:

**Theorem 2** (MLN Hardness). *Given an MLN whose grounded factor graph is $\mathcal{G} = (\mathcal{V}, \mathcal{F})$ in which the weights for interface factors are $w_{G_i} = \log p_i(X)/(1 - p_i(X))$ and constant thresholds $\delta, \{C_i\}_{i \in [n]}$, deciding whether*

$$\forall\{\epsilon_i\}_{i \in [n]} \quad (\forall i. |\epsilon_i| < C_i) \implies |\mathbb{E}R_{MLN}(\{p_i(X)\}_{i \in [n]}) - \mathbb{E}R_{MLN}(\{p_i(X) + \epsilon_i\}_{i \in [n]})| < \delta$$

*is as hard as estimating $\mathbb{E}R_{MLN}(\{p_i(X)\}_{i \in [n]})$ up to $\varepsilon_c$ multiplicative error, with $\epsilon_i = O(\varepsilon_c)$.*

*Proof.* Let $\alpha = [p_i(X)]$, query function $Q(.) = R_{MLN}(.)$ and $\pi_\alpha$ defined by the marginal distribution over interior variables of MLN. Theorem 1 directly implies that $O(1/\varepsilon_c^2)$ queries of a ROBUSTNESS oracle can be used to efficiently estimate $\mathbb{E}R_{MLN}(\{p_i(X)\}_{i \in [n]})$. $\square$

In general, statistical inference in MLNs is #P-complete, and checking robustness for general MLNs is also #P-hard.

# 4 Certifying the Robustness of Sensing-Reasoning Pipeline

Given a sensing-reasoning pipeline with $n$ sensors $\{S_i\}_{i \in [n]}$ and a reasoning component $R$, we will first formally define its end-to-end certified robustness and then its connection to the robustness

of each component. In particular, based on the above hardness result for *certifying the robustness of the reasoning component* in Section 3, we will provide an effective certification method to upper/lower bound the certification, taking *any* oracle for the inference of the reasoning component into account. With the certification of the reasoning component, we will finally provide the robustness certification for the sensing-reasoning pipeline by combining the certification of sensing and reasoning components.

**Definition 3** $((C_I, C_E, p)$-robustness$)$. A sensing-reasoning pipeline with $n$ sensors $\{S_i\}_{i \in [n]}$ and a reasoning component $R$ is $(C_I, C_E, p)$-robust on the input $X$, if for input perturbation $\eta, ||\eta||_p \leq C_I$

$$\left| \mathbb{E}[R(\{p_i(X)\}_{i \in [n]})] - \mathbb{E}[R(\{p_i(X+\eta)\}_{i \in [n]})] \right| \leq C_E.$$

I.e., a perturbation $||\eta||_p < C_I$ on the input only changes the final pipeline output by at most $C_E$.

**Sensing Robustness and Reasoning Robustness.** We decompose the end-to-end certified robustness of the pipeline into two components. The first component, which we call the *sensing robustness*, has been studied by the research community recently [20, 45, 8] — given a perturbation $||\eta||_p < C_I$ on the input $X$, we say each sensor $S_i$ is $(C_I, C_S^{(i)}, p)$-robust if

$$\forall \eta, ||\eta||_p \leq C_I \implies |p_i(X) - p_i(X+\eta)| \leq C_S^{(i)}$$

The robustness of the *reasoning component* R is defined as: Given a perturbation $|\epsilon_i| < C_S^{(i)}$ on the output of each sensor $S_i(X)$, we say the reasoning component $R$ is $\left(\{C_S^{(i)}\}_{i \in [n]}, C_E\right)$-robust if

$$\forall \epsilon_1, ..., \epsilon_n, (\forall i. |\epsilon_i| \leq C_S^{(i)}) \implies \left| \mathbb{E}[R(\{p_i(X)\}_{i \in [n]})] - \mathbb{E}[R(\{p_i(X)+\epsilon_i\}_{i \in [n]})] \right| \leq C_E.$$

It is easy to see that when the sensing component is $\left(C_I, \{C_S^{(i)}\}_{i \in [n]}, p\right)$-robust and the reasoning component is $\left(\{C_S^{(i)}\}_{i \in [n]}, C_E\right)$-robust on $X$, the sensing-reasoning pipeline is $(C_I, C_E, p)$-robust. Since the sensing robustness has been intensively studied by previous work, in this paper, we mainly focus on the reasoning robustness and therefore analyze the robustness of the pipeline.

### 4.1 Certifying Sensing Robustness

There are several existing ways to certify the robustness of sensing models, such as Interval Bound Propagation (IBP) [16], Randomized Smoothing [8], and others [63, 51]. Here we will leverage randomized smoothing to provide an example for certifying the robustness of sensing components.

**Corollary 1.** *Given a sensing model $S_i$, we construct a smoothed sensing model $g_i(X; \hat{\sigma}) = \mathbb{E}_{\xi \sim \mathcal{N}(0, \hat{\sigma}^2)} p_i(X + \xi)$. With input perturbation $||\eta||_2 \leq C_I$, the smoothed sensing model satisfies*

$$\Phi(\Phi^{-1}(g_i(X; \hat{\sigma})) - C_I/\hat{\sigma}) \leq g_i(X + \eta; \hat{\sigma}) \leq \Phi(\Phi^{-1}(g_i(X; \hat{\sigma})) + C_I/\hat{\sigma})$$

*where $\Phi$ is the Gaussian CDF and $\Phi^{-1}$ as its inverse.*
Thus, the output probability of smoothed sensing model can be bounded given input perturbations. Note that the specific ways of certifying sensing robustness is orthogonal to certifying reasoning robustness, and one can plug in different sensing certification strategies.

### 4.2 Certifying Reasoning Robustness

Given the hardness results for certifying reasoning robustness in Section 3.2, in this paper, we assume that we have access to an oracle for statistical inference, and provide a novel algorithm to certify the reasoning robustness. I.e., we assume that we are able to calculate the two partition functions $Z_1(\{p_i(X)\}_{i \in [n]})$ and $Z_2(\{p_i(X)\}_{i \in [n]})$.

**Lemma 4.1** (MLN Robustness). *Given access to partition functions $Z_1(\{p_i(X)\}_{i \in [n]})$ and $Z_2(\{p_i(X)\}_{i \in [n]})$, and maximum perturbations*

---

**Algorithm 1** Algorithms for MLN robustness upper bound (algorithm of lower bound is similar)

---

**input** : Oracles calculating $\widetilde{Z_1}$ and $\widetilde{Z_2}$; maximal perturbations $\{C_i\}_{i \in [n]}$.
**output** : An upper bound for input $R_{MLN}(\{p_i(X) + \epsilon_i\})$

1: $\overline{R}_{min} \leftarrow 1$
2: initialize $\lambda$
3: **for** $b \in$ search budgets **do**
4:    $\lambda \rightarrow \texttt{update}(\{\lambda\}; \lambda_i \in (-\infty, -1] \cup [0, +\infty))$
5:    **for** $i = 1$ **to** $n$ **do**
6:       **if** $\lambda_i \geq 0$ **then**
7:          $\epsilon_i = C_i, \epsilon_i' = -C_i$
8:       **else if** $\lambda_i \leq -1$ **then**
9:          $\epsilon_i = -C_i, \epsilon_i' = C_i$
10:       **end if**
11:       $\overline{R} \leftarrow \widetilde{Z_1}(\{\epsilon_i\}_{i \in [n]}) - \widetilde{Z_2}(\{\epsilon_i'\}_{i \in [n]})$
12:       $\overline{R}_{min} \leftarrow \min(\overline{R}_{min}, \overline{R})$
13:    **end for**
14: **end for**
15: **return** $\overline{R}_{min}$

---

$\{C_i\}_{i\in[n]}$, $\forall \epsilon_1, ..., \epsilon_n$, if $\forall i. |\epsilon_i| < C_i$, we have that
$\forall \lambda_1, ..., \lambda_n \in \mathbb{R}$,

$$\max_{\{|\epsilon_i| < C_i\}} \ln \mathbb{E}[R_{MLN}(\{p_i(X) + \epsilon_i\}_{i\in[n]})] \leq \max_{\{|\epsilon_i| < C_i\}} \widetilde{Z_1}(\{\epsilon_i\}_{i\in[n]}) - \min_{\{|\epsilon_i'| < C_i\}} \widetilde{Z_2}(\{\epsilon_i'\}_{i\in[n]})$$

$$\min_{\{|\epsilon_i| < C_i\}} \ln \mathbb{E}[R_{MLN}(\{p_i(X) + \epsilon_i\}_{i\in[n]})] \geq \min_{\{|\epsilon_i| < C_i\}} \widetilde{Z_1}(\{\epsilon_i\}_{i\in[n]}) - \max_{\{|\epsilon_i'| < C_i\}} \widetilde{Z_2}(\{\epsilon_i'\}_{i\in[n]})$$

*where*
$$\widetilde{Z_r}(\{\epsilon_i\}_{i\in[n]}) = \ln Z_r(\{p_i(X) + \epsilon_i\}_{i\in[n]}) + \sum_i \lambda_i \epsilon_i.$$

We leave the proof to the Appendix B. The high-level proof idea is to decouple $Z_1/Z_2$ into two sub-problems via a collection of Lagrangian multipliers, i.e., $\{\lambda_i\}$. For any assignment of $\{\lambda_i\}$, we obtain a valid upper/lower bound, which reduces the certification process to the process of *searching* for an assignment of these multipliers that minimize the upper bound (maximize the lower bound).

To efficiently search for the optimal assignment of $\{\lambda_i\}$, it is crucial to consider the interactions between these $\{\lambda_i\}$ and the corresponding solution of $\widetilde{Z_r}$, which hinges on the structure of MLN. In particular, we can prove the following (Detailed proofs and discussions in Appendix C):

**Proposition 1** (Monotonicity). *When $\lambda_i \geq 0$, $\widetilde{Z_r}(\{\epsilon_i\}_{i\in[n]})$ monotonically increases w.r.t. $\epsilon_i$; When $\lambda_i \leq -1$, $\widetilde{Z_r}(\{\epsilon_i\}_{i\in[n]})$ monotonically decreases w.r.t. $\epsilon_i$.*

**Proposition 2** (Convexity). *$\widetilde{Z_r}(\{\tilde{\epsilon}_i\}_{i\in[n]})$ is a convex function in $\tilde{\epsilon}_i, \forall i$ with*

$$\tilde{\epsilon}_i = \log \left[ \frac{(1 - p_i(X))(p_i(X) + \epsilon_i)}{p_i(X)(1 - p_i(X) - \epsilon_i)} \right].$$

*Implication.* Given the monotonicity region, the maximal and minimal of $\widetilde{Z_r}$ are achieved at either $\epsilon_i = -C_i$ or $\epsilon_i = C_i$ respectively. Given the convexity region, the maximal is achieved at $\epsilon_i \in \{-C_i, C_i\}$, and the minimal is achieved at $\epsilon_i \in \{-C_i, C_i\}$ or at the zero gradient of $\widetilde{Z_r}(\{\tilde{\epsilon}_i\}_{i\in[n]})$. As a result, our analysis leads to the following certification algorithm.

**Algorithm of Certifying Reasoning Robustness.** Algorithm 1 illustrates the detailed algorithm based on the above result to upper bound the robustness of MLN. The main step is to explore different regimes of the $\{\lambda_i\}$. In this paper, we only explore regimes where $\lambda \in (-\infty, -1] \cup [0, +\infty)$ as this already provides reasonable solutions in our experiments. The function $\texttt{update}(\{\lambda_i\})$ defines the exploration strategy — Depending on the scale of the problem, one can explore $\{\lambda_i\}$ using grid search, random sampling, or even gradient-based methods. For experiments in this paper, we use either grid search or random sampling. It is an exciting future direction to understand other efficient exploration and search strategies. We leave the detailed explanation of the algorithm to Appendix C.

## 5 Experiments

We conduct intensive experiments on five datasets to evaluate the certified robustness of the sensing-reasoning pipeline. We focus on two tasks with different modalities: *image classification* task on Road Sign dataset created based on GTSRB dataset [44] following the standard setting as [17]; and *information extraction* task with stocks news on text data. We also report additional results on two other image classification tasks (Word50 [6] and PrimateNet, which is a subset of ImageNet ILSVRC2012 [9]) with natural knowledge rules in Appendix G and Appendix F. We also report results on standard image benchmarks (MNIST and CIFAR10) with manually constructed knowledge rules in Appendix H. The code is provided at `https://github.com/Sensing-Reasoning/Sensing-Reasoning-Pipeline`.

### 5.1 Experimental Setup

**Datasets and Tasks.** For the *road sign classification* task, we follow [17] and use the same dataset GTSRB [44], which contains 12 types of German road signs {"Stop", "Priority Road", "Yield", "Construction Area", "Keep Right", "Turn Left", "Do not Enter", "No Vihicles", "Speed Limit 20", "Speed Limit 50", "Speed Limit 120", "End of Previous Limitation"}. It consists of 14880 training

samples, 972 validation samples, and 3888 testing samples. We also include 13 additional detectors for knowledge integration, detecting attributes such as whether the border has an octagon shape (See Appendix D for a full list).

For the *information extraction* task, we use the HighTech dataset which consists of both daily closing asset price and financial news from *2006* to *2013* [12]. We choose 9 companies with the most news, resulting in 4810 articles related to 9 stocks filtered by company name. We split the dataset into training and testing days chronologically. We define three information extraction tasks as our sensing models: `StockPrice(Day, Company, Price)`, `StockPriceChange(Day, Company, Percent)`, `StockPriceGain(Day, Company)`. The domain knowledge that we integrate depicts the relationships between these relations (See Appendix E for more details).

**Knowledge Rules.** We integrate different types of knowledge rules for these two applications. We provide the full list of knowledge rules in the Appendix D.

For *road sign classification*, we follow [17], which includes two different types of knowledge rules — *Indication rules* (road sign class $u$ indicates attribute $v$) and *Exclusion rules* (attribute classes $u$ and $v$ with the same general type such as "Shape", "Color", "Digit" or "Content" are naturally exclusive).

For *information extraction*, we integrate knowledge about the relationships between the sensing models (e.g., `StockPrice`, `StockPriceChange`, `StockPriceGain`). For example, the stock prices of two consecutive days, `StockPrice`$(d_1, Company, p_1)$ and `StockPrice`$(d_2, Company, p_2)$, should be consistent with `StockPriceChange`$(d_2, Company, p)$, i.e., $p = (p_2 - p_1)/p_1$.

**Implementation Details.** Throughout the road sign classification experiment, we implement all sensing models using the GTSRB-CNN [13] architecture. During training, we train all sensors with Isotropic Gaussian $\epsilon \sim \mathcal{N}(0, \hat{\sigma}^2 I_d)$ augmented data with 50000 training iterations until converge and tune the training parameters on the validation set, following [8]. We use the SGD-momentum with the initial learning rate as $0.01$ and the weight decay parameter as $10^{-4}$ to train all the sensors for 50000 iterations with 200 as the batch size, following [17]. During certification, we adopt the same smoothing parameter for training to construct the smoothed model based on Monte-Carlo sampling.

For information extraction, we use BERT as our model architecture. During training, we use the final hidden state of the first token [CLS] from BERT as the representation of the whole input and apply dropout with probability $p = 0.5$ on this final hidden state. Additionally, there is a fully connected layer added on top of BERT for classification. To fine-tune the BERT classifiers for three information tasks, we use the Adam optimizer with the initial learning rate as $10^{-5}$ and the weight decay parameter as $10^{-4}$. We train all the sensors for 30 epochs, and the batch size 32.

**Evaluation Metrics.** We adopt the standard *certified accuracy* as our evaluation metric, defined by the percentage of instances that can be certified under certain $\ell_p$-norm bounded perturbations. Specifically, given the input $x$ with ground-truth label $y$, once we can certify the bound of the model's output confidence on predicting label $y$ under the norm-bounded perturbation as $[\mathcal{L}, \mathcal{U}]$, the certified accuracy can be defined by: $\frac{1}{N} \sum_{i=1}^{N} \mathbb{I}([\mathcal{L}_i > 0.5])$ where $\mathbb{I}(\cdot)$ denotes the indicator function. Since each sensing component's certification is performed by randomized smoothing, which yields the failure probability characterized by $\zeta_0$, we will control the failure probability $\zeta$ for the whole sensing-reasoning pipeline pipeline with $n$ sensing models as $\zeta_0 = 1 - (1 - \zeta)^{1/n}$ by applying the union bound. Throughout all the experiments, $\zeta$ is kept to $0.001$ so our end-to-end certification is guaranteed to be correct with at least $99.9\%$ confidence.

## 5.2 Results of *Road Sign Classification*

In this section, we evaluate the certified robustness of our sensing-reasoning pipeline under the $\ell_2$-norm bounded perturbation. We first report the $\ell_2$ certified accuracy of our sensing-reasoning pipeline and compare it to a strong baseline as a vanilla randomized smoothing trained model (without knowledge). Note that it is flexible to replace the sensing component with other robust training algorithms. We conduct our evaluation under different smoothing parameters $\hat{\sigma} = \{0.12, 0.25, 0.50\}$ and various $\ell_2$ perturbation magnitudes on the input image $C_I = \{0.12, 0.25, 0.50, 1.00\}$ (Table 1). During certification, we evaluate our certification time per sample with 25 sensors as 5.39s, which shows that the overall certification time is generally acceptable.

As shown in Table 1, we can see that with knowledge integration, sensing-reasoning pipeline achieves consistently higher certified accuracy compared to the baseline smoothed ML model without

Table 1: **(Road sign classification)** *Certified accuracy* under different input perturbation magnitudes ($C_I$). Models are smoothed with different Gaussian noises $\epsilon \sim \mathcal{N}(0, \hat{\sigma}^2 I_d), \hat{\sigma} \in \{0.12, 0.25, 0.50\}$. Rows with $*$ denote the best certified accuracy among all the smoothing parameters for each method. The bold numbers show the higher certified accuracy under the same ($C_I, \hat{\sigma}$) setting and the numbers with underline show the highest certified accuracy for each $C_I$ among different smoothing parameters. (All certificates hold with $p = 99.9\%$)

| Methods | $\hat{\sigma}$ | $C_I = 0.12$ | $C_I = 0.25$ | $C_I = 0.50$ | $C_I = 1.00$ |
|---|---|---|---|---|---|
| Vanilla Smoothing (w/o knowledge) | 0.12 | 90.8 | 87.1 | 0.0 | 0.0 |
| | 0.25 | 89.6 | 88.4 | 71.6 | 0.0 |
| | 0.50 | 84.0 | 80.2 | 73.2 | 61.7 |
| | $*$ | 90.8 | 88.4 | 73.2 | 61.7 |
| Sensing-Reasoning Pipeline (w/ knowledge) | 0.12 | **96.0** | **89.0** | **73.2** | **24.2** |
| | 0.25 | **93.4** | **91.0** | **74.0** | **49.2** |
| | 0.50 | **89.3** | **85.4** | **75.5** | **62.5** |
| | $*$ | **96.0** | **91.0** | **75.5** | **62.5** |

knowledge under all the perturbation magnitudes $C_I$ and smoothing parameter $\hat{\sigma}$ settings. Under the small perturbation magnitude cases, our improvement is very significant (around $5\%$). More interestingly, given large $C_I$ but small smoothing parameter $\hat{\sigma}$, vanilla randomized smoothing-based certification directly fails ($0\%$ certified accuracy) due to the looseness of the hypothesis testing bound, while the sensing-reasoning pipeline could still achieve reasonable certified robustness (over $71\%$ on $C_I = 0.50$, $49\%$ on $C_I = 1.00$) under the same ($C_I, \hat{\sigma}$) settings. This indicates a very realistic case: we always **under-estimate** the attacker's ability easily under the real-world setting – in this case, the sensing-reasoning pipeline could remain robust even provide reasonable certified accuracy with a conservative smoothing parameter.

### 5.3 Results of *Information Extraction*

In this section, we conduct the certified robustness evaluation on the information extraction task on text data. Since there is no good certification method on discrete NLP data for sensing models, we directly assume the maximal perturbation on the output of sensors ($C_S$). Table 2 shows the certified accuracy on the final outputs of the reasoning component. We see that the sensing-reasoning pipeline provides

Table 2: **(Information extraction)** *Certified accuracy* under different perturbation magnitudes ($C_S$) based on the sensing models' output uncertainty. (All certificates hold with 99.9% confidence)

| Methods | $C_S = 0.1$ | $C_S = 0.5$ | $C_S = 0.9$ |
|---|---|---|---|
| Vanilla Smoothing (w/o knowledge) | 99.7 | 94.7 | 38.4 |
| Sensing-Reasoning Pipeline (w/ knowledge) | **100.0** | **100.0** | **58.8** |

significantly higher certified robustness, and even under a high perturbation magnitude on all sensing models' output confidence ($C_S = 0.5$), which means the sensing-reasoning pipeline can still leverage the knowledge to help enhance the robustness given strong attacker. To further illustrate intuitively why such knowledge-based reasoning helps, Figure 3 shows the "margin" — the probability of the ground truth class minus the probability of the wrong class — with or without knowledge integration. We see that, with knowledge integration, we can significantly increase the number of examples with a large "margin" under adversarial perturbations. This explains the improvement of certified robustness, which highly relies on such prediction confident margin.

We also conduct experiments on PrimateNet, Word50, MNIST, CIFAR10 datasets for the image classification tasks in Appendix F- Appendix H. We observe similar results that knowledge integration significantly boosts the certified robustness.

## 6    Related Work

**Robustness for Single ML model and ML Ensemble.** Lots of efforts have been made to improve the robustness of single ML or ensemble models. Adversarial training [15], and its variations [47, 31, 53] have generally been more successful in practice, but usually come at the cost of accuracy and increased training time [48, 53]. To further provide certifiable robustness guarantees for ML models, various certifiable defenses and robustness verification approaches have been proposed [20, 45, 8, 27, 25]. Among these strategies, randomized smoothing [8] has achieved scalable performance. With improvements in training, including pretraining and adversarial training, the certified robustness

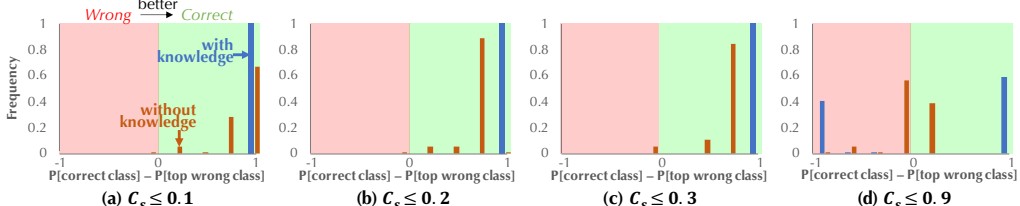

Figure 3: **(Information extraction)** Histogram of the **robustness margin** (the difference between the probability of the correct class (lower bound) and the top wrong class (upper bound)) under perturbations. If such a difference is positive, it means that the classifier makes the right prediction under perturbations.

bound can be further improved [4, 42]. In addition to the single ML model, some work proposed to promote the diversity of classifiers and therefore develop a robust ML ensemble [34, 59, 57, 58]. Although promising, these defense approaches, either empirical or theoretical, can only improve the robustness of a single ML or ensemble model. Certifying or improving the robustness of such single or pure ensemble models is very challenging, given that there is no additional information that can be utilized. In addition, the ML learning process usually favors a pipeline that is able to incorporate different sensing components as well as domain knowledge in practice. Thus, certifying the robustness of such pipelines is of great importance.

**Robustness of End-to-end ML Systems.** There have been intensive studies on joint inference between multiple models, and the predictions based on joint inference can help to further improve the clean accuracy of ML pipelines [55, 10, 38, 33, 7, 5], which have been applied to a range of real-world applications [2, 37, 32]. Often, these approaches use different statistical inference models such as factor graphs [50], Markov logic networks [41], and Bayesian networks [35] as a way to integrate domain knowledge. In this paper, we take a different perspective on this problem — instead of treating joint inference as a way to improve the *clean accuracy*, we explore the possibility of using it as exogenous information to improve the end-to-end *certified robustness* of ML pipelines. A recent work [17] explores the empirical robustness improvement via knowledge integration, while there is no robustness guarantee provided. As we show in this paper, by integrating domain knowledge, we are able to improve the *certified robustness* of the ML pipelines significantly.

## 7   Conclusions

We provide the first certifiably robust sensing-reasoning pipeline with knowledge-based logical reasoning. We theoretically prove the certified robustness of such ML pipelines, and provide complexity analysis for certifying the reasoning component. Our extensive empirical results demonstrate the certified robustness of sensing-reasoning pipeline, and we believe our work would shed light on future research towards improving and certifying robustness for general ML frameworks as well as different ways to integrate logical reasoning with statistical learning.

**Acknowledgements**  This work is partially supported by the NSF grant No.1910100, NSF CNS No.2046726, C3 AI, and the Alfred P. Sloan Foundation. CZ and the DS3Lab gratefully acknowledge the support from the Swiss State Secretariat for Education, Research and Innovation (SERI) under contract number MB22.00036 (for European Research Council (ERC) Starting Grant TRIDENT 101042665), the Swiss National Science Foundation (Project Number 200021_184628, and 197485), Innosuisse/SNF BRIDGE Discovery (Project Number 40B2-0_187132), European Union Horizon 2020 Research and Innovation Programme (DAPHNE, 957407), Botnar Research Centre for Child Health, Swiss Data Science Center, Alibaba, Cisco, eBay, Google Focused Research Awards, Kuaishou Inc., Oracle Labs, Zurich Insurance, and the Department of Computer Science at ETH Zurich. HG has received funding from the European Research Council (ERC) under the European Union's Horizon 2020 research and innovation programme (grant agreement No. 947778).

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
