# OpenReview forum: "Improving Certified Robustness via Statistical Learning with Logical Reasoning"
_NeurIPS.cc/2022/Conference — NeurIPS 2022 Accept_

### Official Review · Reviewer_ApnS · 2022-07-07

**Rating:** 8
**Confidence:** 1
**Soundness:** 3 good
**Presentation:** 3 good
**Contribution:** 3 good

**Summary:**

This paper proposed to integrate statistical ML models with knowledge as a reasoning component using Markov logic networks in order to improve certified robustness. They start from a sensing-reasoning pipeline, followed by deriving the hardness of certifying general reasoning model, and finally certifying the robustness of sensing-reasoning pipeline. Experiments on five datasets show that their method outperforms models from prior work.

**Questions:**

N/A

**Ethics Review Area:**

["I don’t know"]

**Limitations:**

Same as the weakness point mentioned in the above section.

**Strengths And Weaknesses:**

I'm not an expert in this area so I didn't check the correctness of the math thoroughly.

Overall I think this is a nice paper with thorough theoretical analysis and solid experiments. The paper is well written.

One weakness that I can think of is that it would be great if you could provide more experiments on large-scale datasets.

---

> ### Author Response · Authors · 2022-08-02
> **Response to Reviewer ApnS**
>
> Thanks for your appreciation of our work. We address your concerns below.
>
> > it would be great if you could provide more experiments on large-scale datasets.
>
> Thanks for your valuable suggestion. In our paper, we have evaluated our pipeline on the PrimateNet dataset, which is a subset of the scalable ImageNet ILSVRC2012, as one of the large-scale datasets we could use. In addition, we have evaluated the Road sign classification task which contains 14880 real-world training instances with high resolution as a large-scale dataset. We have evaluated our pipeline on the Word50 dataset which is a standard image dataset for both word and character level classification. We have also evaluated the NLP task with Internet-scale collected stock news for relation extraction.
>
> Moreover, following the suggestions from reviewers, we have added additional experimental results on standard tasks such as MNIST and CIFAR10 to demonstrate SOTA results. So far, we have evaluated our pipeline on ImageNet scale image tasks and NLP tasks, and we believe further exploring other large-scale datasets with useful knowledge rules could be an interesting future direction.

---

### Official Review · Reviewer_ZvBd · 2022-07-11

**Rating:** 5
**Confidence:** 3
**Soundness:** 2 fair
**Presentation:** 2 fair
**Contribution:** 2 fair

**Summary:**

This paper tries to introduce certified robustness into a probabilistic reasoning system. First, Markov logic network (MLN) is used as a generic model that can represent every probabilistic logical reasoning system. Then, the authors proved that the complexity of certifying a MLN model reasoning model robustness is #P-Hard. Furthermore, a novel algorithm to certify the reasoning robustness is proposed with an assumption that two partition functions are available. Experimental results on both road sign classification and information extraction show that sensing-reasoning pipeline can have much higher certified accuracy. Detailed analysis showed that by incorporating reasoning components, the number of examples with a large “margin” under adversarial attacks significantly increased.

**Questions:**

1.	Is there a clear definition of \textit{certified robustness}? The paper should include this definition for self-completion.
2.	Some notions are used but not defined. For example, at line#154, what is $f_H()$?
3.	In section 4.2, the partition functions $Z_1$ and %Z_2$ are assumed to be available beforehand. How are these functions obtained for the two tasks in the experiments?
4. What is the key point the authors want a reader should take away from this paper? I am confused by the independent points in this paper.

**Ethics Review Area:**

["I don’t know"]

**Limitations:**

Certified robustness is an important topic that can be beneficial to societal good through building robust and trustworthy machine learning models.

**Strengths And Weaknesses:**

Strengths: \
1.	Certified robustness is a very important research topic.\
2.	Theoretical analysis in this paper is valuable.

Weaknesses:\
It is not clear what the key idea the paper wants to deliver is. There are several points in this paper: \
a. the complexity of certifying robustness; \
b. an algorithm to estimate the upper/lower bound; \
c. experimental results show that incorporating reasoning components can improve the certified robustness. \
However, it seems they are all independent between each other. I cannot see a coherent story line in this paper. For example, experimental results show that incorporating reasoning components can significantly improve robustness. But, how are these results related to the proposed algorithm on upper/lower bound estimation or the complexity?

---

> ### Author Response · Authors · 2022-08-02
> **Response to Reviewer ZvBd**
>
> Thanks for your constructive comments on this work. Here are the answers to your major concerns.
>
> > It is not clear what the key idea the paper wants to deliver is …  I cannot see a coherent story line in this paper.
>
> Thanks for the comment. We are glad that the reviewer pointed out the three points (**a.** the complexity of certifying robustness; **b.** an algorithm to estimate the upper/lower bound; **c.** experimental results show that incorporating reasoning components can improve the certified robustness), and they are fully coherent and dependent: We first provide the complexity analysis on certifying the robustness for a reasoning component such as MLN and prove that it is #P-complete (a). Given such harness analysis, we thus proposed the concrete certification algorithm to estimate the upper/lower bound of the prediction probability of the reasoning component, which serves as the certified robustness of the reasoning component by definition (b). Finally, by combining our certification algorithm for the reasoning component with existing certification algorithms for sensing models, we are able to provide the end-to-end robustness certification of our sensing-reasoning pipeline and evaluate our certified robustness on various tasks (c).
>
>
> > Is there a clear definition of \textit{certified robustness}? The paper should include this definition for self-completion.
>
> Thanks for the comment. We follow the standard definition of certified robustness: the lower bound of model prediction accuracy given inputs considering an adversarial perturbation with bounded magnitude [1] (the perturbation magnitude in our paper is bounded by the l2 norm, denoted as radius C_I). We have added the definition in our revision line 171-174 with reference. We have also provided the formal definition of certified robustness considering the sensing-reasoning pipeline in Definition 3.
>
> [1] Cohen, J., Rosenfeld, E., & Kolter, Z. (2019, May). Certified adversarial robustness via randomized smoothing. In International Conference on Machine Learning (pp. 1310-1320). PMLR.
>
> > Some notions are used but not defined. For example, at line#154, what is $f_H()$?
>
> Thanks for pointing it out.  $f_H()$ means the factor function w.r.t the interior factor $H \in \mathcal{H}$ (line 138-143).
>
> > In section 4.2, the partition functions $Z_1$ and $Z_2$ are assumed to be available beforehand. How are these functions obtained for the two tasks in the experiments?
>
> Thanks for the comment. The formal definition of partition functions $Z_1$ and $Z_2$ are in Section 2.2 (line 154). In experiments, we compute $Z_1$ and $Z_2$ by iterating all the possible worlds and adding the weights of each possible world (related to the weights of each sensing model p_i and weights of knowledge rules) correspondingly. Implementation details can be found in Appendix D-G and our code repo for each application.
>
> > What is the key point the authors want a reader should take away from this paper?
>
> Thanks for the insightful question.
> The key idea we aim to convey is that: with domain knowledge and reasoning ability integration, the certified robustness of machine learning models/pipelines can be significantly improved. Although this intuitively makes sense, such a pipeline design and robustness certification are challenging and interesting. To show this, we start by designing the whole sensing-reasoning framework, analyzing the complexity of certifying the reasoning component, proposing the detailed certification algorithm for the reasoning component and thus the end-to-end pipeline, and finally showing the state-of-the-art certified robustness results on different tasks.
>
> Moreover, we want to note that as we discussed above, the main contributions of the paper **1)** hardness results of certifying the robustness of the reasoning component, **2)** the algorithm to estimate the upper/lower bound for certifying the reasoning component, and **3)** robustness certification results on different tasks, are dependent on each other as one coherent story, which is also appreciated by reviewer g8Bp. We will make it more clear in our revision and thanks for the valuable suggestion!

---

> > ### Comment · Reviewer_ZvBd · 2022-08-03
> > **comments to author response**
> >
> > Thanks to the response!
> > To my question of "coherent story line", the author's answer is not satisfying.  "Given such harness analysis, we thus proposed...". In fact, without the complexity analysis, the proposed estimation algorithm still valid, because the deriving process of the algorithm does not depend on the complexity analysis at all. Given the assumption that the partition functions are known, the complexity of the algorithm is not related to the complexity analysis in the previous section. Therefore, the complexity analysis in the previous section does not contribute anything to the algorithm design.
> >
> > The authors said "we compute and by iterating all the possible worlds and adding the weights of each possible world...". However, the complexity of iterating all possible worlds may be very high as the dimensions of logic space increases. Does this mean the proposed estimation algorithm is still  #P-complete? If so, this will significantly degrade the contribution of this paper.

---

> > > ### Author Response · Authors · 2022-08-03
> > > **Response**
> > >
> > > > Q1. To my question of "coherent story line", the author's answer is not satisfying.
> > >
> > > Thank you for the feedback and we really appreciate it!! Sorry for the confusion. By saying  "Given such harness analysis, we thus proposed...", we meant that given the proved #p-completness of certifying the robustness of the reasoning component, we therefore know that it is impossible to obtain a polynomial time algorithm for the certification problem; and thus we need to either make additional assumptions to solve the problem or use certain heuristics. As a result, we make the assumption on oracle access to the inference of Z1 and Z2, and carefully partition the space of coefficient $\lambda$ to provide the upper/lower bound for the certification in polynomial time as shown in Algorithm 1. In particular, we are able to prove that under certain regions of $\lambda$, the certification problem would be either monotonic or convex and therefore we can provide the robustness certification under each region, which is one of our main contributions. That is to say, although the hardness results do not directly lead to the certification algorithm itself, it provides clear guidance on what principles we should follow to design the algorithm: just like the contribution of any other hardness analysis in algorithmic research, which tells whether we should work hard to find a polynomial algorithm or it is infeasible so we should consider other heuristics or additional assumptions. We believe that the hardness analysis, together with the algorithm design, makes this work more rigorous. We hope the logic is clear, and essentially it is the same motivation as any other hardness studies which motivate the follow-up algorithm design. We are happy to further clarify it if there are other questions.
> > >
> > >
> > > > Q2. The authors said "we compute and by iterating all the possible worlds and adding the weights of each possible world..."
> > >
> > > Thanks for the questions about the complexity of inference for Z1 and Z2. We note that our paper mainly focuses on providing the robustness certification for the reasoning component (which is proved to be #p-complete) given any inference oracle of Z1 and Z2, and our proposed algorithm 1 will provide an upper/lower bound of the certificate in polynomial time given the oracle, which is one of our contributions.
> > >
> > > In addition, the complexity of the MLN inference itself is independent of our contribution. Actually, there have been decades of research on efficient (approximate) algorithms for MLN inference, which is still an active research area itself. As long as these algorithms provide an inference upper/lower bound that can serve as an inference oracle for Z1 and Z2, we can directly plug them into our pipeline for the robustness certification. In our paper, we directly used the most generic oracle as enumerating all possible worlds for convenience, and we will discuss other possible inference oracles along this line of research in our revision.
> > >
> > > Thanks for the follow-up questions and we will add these discussions in our revision! We look forward to discussions with the reviewer to help further improve the quality of our work.

---

> > > > ### Comment · Reviewer_ZvBd · 2022-08-05
> > > > **comments to author response (round 2)**
> > > >
> > > > Thanks for the further clarifications.
> > > > For Q1, I accept the author's explanation. But, at the same time, I would like to insist that the authors need to improve the paper writing to reflect the logic flow mentioned in the author response and make the story more coherent. I will update my evaluation score accordingly.

---

> > > > > ### Author Response · Authors · 2022-08-07
> > > > > **Thanks for you feedback!**
> > > > >
> > > > > Thank you so much for your suggestions and comments and we will definitely improve our paper to make the logic more clear. In particular, following the suggestions, we have added a roadmap at the end of the introduction (line 93-99) in our revision, and we have emphasized the relationship between our hardness analysis and the proposed certification algorithm (line 175-178, line 222-227).
> > > > >
> > > > > Thank you for the helpful discussion for improving our work again! Please let us know if you have more feedback and we will try our best to fix them!

---

### Official Review · Reviewer_7FcD · 2022-07-13

**Rating:** 6
**Confidence:** 3
**Soundness:** 2 fair
**Presentation:** 3 good
**Contribution:** 3 good

**Summary:**

This paper proposes to improve robustness certification by leveraging a reasoning component based on Markov logic networks (MLN). Human knowledge (i.e., logical rules) are formalized as variables and factors in the reasoning component, thereby guiding the sensing-reasoning pipeline to exhibit better certified robustness. Concretely, the contribution of this paper is three-fold:

1. This paper proves that the computational complexity of certifying the robustness of MLN is #P-hard. The basic idea is to reduce the famous #P-complete COUNTING problem to checking for reasoning robustness. This hardness result tells us why this task is challenging, indicating that the key points to mitigate it are: (1) choosing a "good" reasoning component that have efficient inference algorithm; (2) find an efficient approximation algorithm that can reduce the problem of cerifying the robustness to the statistical inference oracle, and prove how good (the bounds) the algorithm is.
2. This paper complete the two key points mentioned above. The reasoning component is designed as Markov logic networks (MLN), of which the inference is #P-complete but has efficient approximation algorithms that can be regarded as oracles. Based on this, the authors provide an effective certification method for reasoning robustness.
3. Experimental results on several applications demonstrate the robustness of such ML pipelines.

**Questions:**

Q1: maybe the authors can provide an explicit definition of "Statistical Inference" as background knowledge. In Figure 1, it looks like statistical inference is done by the sensing component, which is decoupled with the reasoning component. However, in Section 3 and 4, we say "statistical inference of the reasoning component".

Q2: In Section 4, the authors use the wording "we assume that we have access to an oracle for statistical inference". I think it could be better if the authors can explicitly tell us what the concrete oracles are.

Q3: L86 "Surprisingly, providing certified robustness for MLN is possible because of the structure inherent in the probabilistic graphical models and distributions in the exponential family." Maybe the authors can explain more about this point as it is still unclear for me how this point is concluded from Section 4.

**Limitations:**

Q4: (about the limitation) what is the limitation of MLN comparing to other statistical inference models? Can the methodology proposed in this paper generalize to others?

**Strengths And Weaknesses:**

Pros:
- Certified robustness is an important topic in machine learning, and it is promising that introducing human knowledge expressed as logical rules can further improve the overall certified robustness. This topic is challenging, as discussed in Section 3.
- Knowledge is expressed in the framework of Markov logic networks. This is elegant and easy-to-understand. This design choice also helps the authors propose the efficient algorithm.
- Key points in this paper are well formulated and proven.
- Good experimental results on different applications, with good result analysis.

Cons:
- It is quite hard for me to understand the relationships among Section 2, 3, and 4 (especially how Section 4 is based on Section 3), as I'm not an expert in this area. I try to summarize my understanding in the "Summary" part above, but I'm not sure whether it is correct or not. I hope that the authors can point out incorrect understandings in my summary in the rebuttal. Moreover, I also list detailed questions that are still unclear for me as follows.

---

> ### Author Response · Authors · 2022-08-02
> **Response to Reviewer 7FcD**
>
> We thank the reviewer’s appreciation for our work, and we address the questions below.
>
> > It is quite hard for me to understand the relationships among Section 2, 3, and 4 (especially how Section 4 is based on Section 3).
>
> Sorry for the confusion. In Section 2, we introduced our sensing-reasoning pipeline (Section 2.1) and illustrated the formal definition and technical details of the reasoning component using MLN (Section 2.2). Then, we aim to provide the robustness certification for such a pipeline, which leads to the main challenge as how to provide the certification for an MLN, and this requires us to answer two questions: **(Q1)** what is the computational complexity of providing certification for an MLN? and **(Q2)** given a rigorous answer for Q1, how can we design a concrete certification algorithm? This structure follows the standard algorithmic research to first understand the hardness of the problem that we try to solve, and then provide the algorithm based on its hardness results (e.g., if the problem is in P, we should provide a polynomial algorithm to it).
>
> Thus, in Section 3, we first discussed the hardness of certifying a general reasoning model (Section 3.1), and extended it to a specific reasoning model MLN (Section 3.2), proving that certifying the robustness of the MLN program is #P-complete.
>
> As a result, based on our hardness result in section 3, we know that the robustness certification of MLN is #P-complete, and thus in Section 4, we propose an oracle assumption for MLN inference and accordingly provide a novel robustness certification algorithm for MLN, which is one of our main contributions. Putting the existing certification for sensing models and our approach for the reasoning component (i.e., MLN in our paper) together, we are finally able to provide the end-to-end certification for the proposed sensing-reasoning pipeline, followed by experimental evaluations in section 5.
>
> > maybe the authors can provide an explicit definition of "Statistical Inference" as background knowledge. In Figure 1, it looks like statistical inference is done by the sensing component … in Section 3 and 4, we say "statistical inference of the reasoning component".
>
> Thanks for pointing this out and sorry for the confusion. It should have been “Statistical Learning” in Figure 1, specifically referring to sensing models learned as neural networks. We use the inference result of these neural networks as the input to the reasoning component. By “statistical inference of the reasoning component”, we mean the inference process of Markov logic networks (we further clarified it in Line 153). We have updated Figure 1 for clarification in our revision.
>
> > it could be better if the authors can explicitly tell us what the concrete oracles are.
>
> Thanks for the comment. The oracle here means $\tilde{Z}$ as defined in Line 258, which is an oracle for the inference of the partition function $Z$.
>
> > L86 "Surprisingly, providing certified robustness for MLN is possible because of the structure inherent in the probabilistic graphical models and distributions in the exponential family." Maybe the authors can explain more about this point as it is still unclear for me how this point is concluded from Section 4.
>
> Thanks for your insightful comments. First, as shown in Lemma 4.1, the robustness of MLN is provided by the estimation of the lower and upper bound of $\widetilde{Z_r}(\{\epsilon_{i\, i \in [n]}})$, while this calculation is based on the properties “monotonicity” and the “convexity” under certain conditions as stated in Line 268. Concretely, “the structure inherent in the probabilistic graphical models and distributions in the exponential family” means the monotonicity and convexity properties that make solving the min-max problem for certifying robustness possible as shown in Appendix C. Since the proof is deferred to the appendix, it is indeed a bit unclear to see this point, and we will move some content from the appendix back to the main paper and explain it more clear in our revision. Thanks for the valuable suggestions.
>
>
> > what is the limitation of MLN comparing to other statistical inference models? Can the methodology proposed in this paper generalize to others?
>
> One limitation of MLN is the #P-complete complexity during the inference, which would lead to high computation complexity when the number of sensing models is large.
> Thanks for the question about generalization. Actually, our pipeline can be generalized to other reasoning models such as the Bayesian Network (BN). We have added the hardness analysis of certifying BN and the concrete BN certification algorithm in Appendix J in our revision following the suggestions.

---

> > ### Comment · Reviewer_7FcD · 2022-08-03
> > **The author response does not explain the oracle directly**
> >
> > Thanks to the response, but the answer to Q2 is not satisfying.
> >
> > - Q2: it could be better if the authors can explicitly tell us what the concrete oracles are.
> >   - I'm not asking how the oracle is notated. Insteadly, I want to know what it is and how it is implemented in the experiments. In this paper, the authors just mention that "we assume that there are two partition functions Z1 and Z2. They are oracles.", without any details.
> >   - I totally understand that, in complexity theory, an oracle is a black-box machine that can solve the specified problem in O(1), of which the implementations are not important. However, in my understanding, oracles are used to understand the complexity of problems. If we want to propose an algorithm based on some oracle, I think at least one of the following two conditions should be satisfied: (1) there is an efficient (approximate) solver for the oracle problem; (2) the oracle is expected to be run on non-Turing machines (e.g., quantum computers). Otherwise, I don't think it reasonable to provide algorithms with black-box oracles.
> >
> > This is a serious problem. Therefore, I lower the rating unless this problem can be well addressed.

---

> > > ### Author Response · Authors · 2022-08-03
> > > **Concrete implementation of the oracle**
> > >
> > > Thanks for your feedback, and we should have definitely explained this more clear – sorry we misunderstood the question before as asking for a concrete formal definition of the oracle.
> > > We definitely agree that it is crucial to make it clear how the inference oracle of Z1 and Z2 are implemented in practice, and we provide the implementation details and related discussion of the oracle below.
> > >
> > > For general MLN structure as the reasoning component discussed in our paper, we compute the inference oracle by enumerating all possible worlds, which is feasible when # variables is relatively small. The concrete computation code is in the “certify.py” under different application folders (e.g., L93-102 under the Word50-10 folder).
> > >
> > > We note that our paper mainly focuses on providing the robustness certification for the reasoning component (which is proved to be #p-complete) given **any** inference oracle, and our proposed algorithm 1 will provide an upper/lower bound of the certificate. Thus, in our paper, we directly used the most generic oracle as enumerating all possible worlds, but there are other possibilities for such inference oracle as we will discuss below.
> > >
> > > In particular, for reasoning component that is of special structures, such as a tree structure MLN, oracle of Z1 and Z2 can be computed in polynomial time, using algorithms such as variable elimination [1].
> > > In our work, we aim to develop a generic sensing-reasoning pipeline, and thus we didn’t consider the special MLN structures here. But these efficient algorithms can be directly plugged into our pipeline.
> > >
> > > Moreover, for even larger scale problems without these structures, we could also use approximate algorithms to estimate the inference oracle for Z1 and Z2 — there is a long line of research about approximate inference for MLN (or graphical model in general) including MCMC, lifted inference, and variational inference [2,3]. In principle, any approximate algorithm that returns a upper/lower bound of Z1 and Z2 can be plugged into our pipeline as an oracle.
> > >
> > > We will revise our paper to make it clear about our current procedure of computing the inference oracle for Z1 and Z2. We will also add a discussion of future research directions which can take advantage of these approximate inference algorithms.
> > >
> > > We really appreciate the feedback from you! Please let us know if you have any other questions or suggestions!
> > >
> > >
> > >
> > > [1] Dechter, Rina. "Bucket elimination: A unifying framework for probabilistic inference." Learning in graphical models. Springer, Dordrecht, 1998. 75-104.
> > >
> > > [2] Deepak Venugopal, Vibhav G. Gogate. Scaling-up Importance Sampling for Markov Logic Networks. NIPS 2014
> > >
> > > [3] Niepert, Mathias, and Guy Van den Broeck. "Tractability through exchangeability: A new perspective on efficient probabilistic inference." AAAI 2014.

---

> > > > ### Comment · Reviewer_7FcD · 2022-08-04
> > > > **My concern is addressed**
> > > >
> > > > Thanks to your clarification and it addresses my concern.
> > > >
> > > > I update the rating to 6.

---

### Official Review · Reviewer_g8Bp · 2022-07-14

**Rating:** 6
**Confidence:** 3
**Soundness:** 3 good
**Presentation:** 3 good
**Contribution:** 3 good

**Summary:**

This paper investigates a couple different things, but at its core its investigating the certified robustness of systems that leverage "logical reasoning" (i.e. Markov Logic Networks) on top of regular statistical learning neural networks (what this paper calls a "sensing-reasoning" pipeline).

This paper has 3 primary contributions:
1. It examines the problem of certifying robustness for graphical models as well as its computational complexity.
2. It proposes an approach for certifying robustness for these models.
3. It demonstrates that on a variety of tasks it looks at, this sensing + reasoning pipeline can lead to better robustness than pure sensing.



**Questions:**

One thing that was confusing to me was what the baseline "sensing only" models actually *were*? For example, on the PrimateNet tasks, there are a bunch of "leaf sensing" nodes that the reasoning component leverage to make its final prediction. However, how is the model without the "reasoning" actually arriving at its final prediction?

**Limitations:**

Yes.

**Strengths And Weaknesses:**

Strengths:
1. I hadn't thought about this particular angle before, but I think it's a very interesting and relevant problem to tackle. One pitch for hybrid neuro-symbolic approaches for quite some time has been robustness, but I haven't seen any papers tackle certified robustness for these systems before, which made their claims of "robustness" less convincing.
2. The proof of the difficulty of certifying robustness on the reasoning component seems useful, although I lack the expertise to verify how interesting it is (the reduction wasn't obvious to me at least).
3. The approach for certifying robustness of the reasoning component seems reasonable and fairly interesting to me.
4. The authors demonstrate that their sensing+reasoning pipeline achieves better certified robustness than pure sensing approaches on a wide array of datasets/tasks, which is great!
5. I also appreciated the intuitive examples of why sensing + reasoning would lead to better robustness compared to pure sensing.

Weaknesses:
The main issue I have about this paper relate to the specific tasks it uses, which is related to the specific kinds of "sensing + reasoning" networks it uses. As a result of the particular kind of pipeline it requires, the tasks it evaluates on are not standard "certified robustness" tasks, as far as I can tell. This means that they do not compare against any existing papers/baselines on their tasks, only the baselines they've created for these tasks. This is somewhat worrisome, although the baselines they choose do seem reasonable.

It would be much  more convincing to me if the authors were able to demonstrate SOTA robustness results against standard robust baselines.

Overall, the authors do a great job of introducing the task of certified robustness to a new class of (quite interesting!) models, examine its difficult and computational complexity, and evaluating its robustness on a wide array of tasks. However, the lack of experiments against standard established baselines and tasks gives me some pause. Still, I think this paper is quite interesting and should be accepted.

---

> ### Author Response · Authors · 2022-08-02
> **Response to Reviewer g8Bp (1/2)**
>
> Thanks for your insightful comments and suggestions. We provide our answers to the questions below, and we have updated our revision following the suggestions.
>
> >It would be much more convincing to me if the authors were able to demonstrate SOTA robustness results against standard robust baselines.
>
> We thank the reviewer for this insightful comment. Note that in addition to the robustness brought by knowledge and logical inference of our sensing-reasoning pipeline, it is also independent with the training methods for the sensing part. In other words, any SOTA robust training methods like SmoothAdv[1] and Consistency[2] can be easily applied in our pipeline to further boost the overall robustness.
>
> Following the suggestion, we conduct additional experiments to compare with other SOTA certified robustness baselines on standard tasks: we evaluated the certified robustness of our sensing-reasoning pipeline on MNIST and CIFAR10 datasets based on constructed knowledge rules. We show that our sensing-reasoning pipeline outperforms the SOTA certified robustness baselines on these standard tasks, especially under large radii. The detailed settings and results are shown as follows:
>
> **MNIST**:
>
> Knowledge rules: We construct five attributes and randomly assign them to four different digits, so that each digit will exactly contain two attributes. We build the indication rules between each attribute and its corresponding digits, and the exclusion rules between different digit classes.
>
> Implementation details: All the models are trained with SOTA baseline Consistency[2], with the consistency hyperparameter $\lambda$ as 5. During certification, we adopt the same smoothing parameter for training to construct the smoothed model based on Monte-Carlo sampling.
>
> |                  Methods                 | $C_I=0.00$ | $C_I=0.25$ | $C_I=0.50$ | $C_I=0.75$ | $C_I=1.00$ | $C_I=1.25$ | $C_I=1.50$ | $C_I=1.75$ | $C_I=2.00$ |
> |:----------------------------------------:|:----------:|:----------:|:----------:|:----------:|:----------:|:----------:|:----------:|:----------:|:----------:|
> |                Consistency               |    99.5    |    98.9    |    **98.0**    |    96.0    |    93.0    |    87.8    |    78.5    |    60.5    |    41.7    |
> | Sensing-Reasoning  |  **99.6**  |  **98.2**  |    97.6    |  **96.3**  |  **93.5**  |  **88.2**  |  **78.9**  |  **61.2**  |  **43.2**  |
>
> **CIFAR10**:
>
> Knowledge rules: We randomly generate ten attributes for CIFAR10, and for each attribute, it will be randomly assigned to 3 to 7 different categories. We build the indication rules between each attribute and its corresponding classes, and the exclusion rules between different classes.
>
> Implementation details: All the models are trained with SOTA, Consistency[2], with the consistency hyperparameter $\lambda$ as 10. During certification, we adopt the same smoothing parameter for training to construct the smoothed model based on Monte-Carlo sampling.
>
> |                  Methods                 | $C_I=0.00$ | $C_I=0.25$ | $C_I=0.50$ | $C_I=0.75$ | $C_I=1.00$ | $C_I=1.25$ | $C_I=1.50$ | $C_I=1.75$ | $C_I=2.00$ |
> |:----------------------------------------:|:----------:|:----------:|:----------:|:----------:|:----------:|:----------:|:----------:|:----------:|:----------:|
> |       Consistency        |  77.8  |  68.8  |  **57.4**  |  43.8  | 36.2  |  29.5  |  22.9  |  19.7  | 16.6 |
> | Sensing-Reasoning| **78.4** | **70.4** |  56.2  | **46.0** | **37.4** | **29.6** | **25.2** | **21.8** | **18.8** |
>
> From the results, we can see that the sensing-reasoning pipeline outperforms the SOTA Consistency baseline in terms of the certified robustness even with the simple knowledge rules. Generally, we should expect higher certified robustness by integrating with natural and meaningful knowledge rules (e.g., road sign classification and information extraction tasks as shown in our paper). We also updated these new results in Appendix H in our revised version.
>
> [1] Salman, H., Li, J., Razenshteyn, I., Zhang, P., Zhang, H., Bubeck, S., & Yang, G. (2019). Provably robust deep learning via adversarially trained smoothed classifiers. Advances in Neural Information Processing Systems, 32.
>
> [2] Jeong, J., & Shin, J. (2020). Consistency regularization for certified robustness of smoothed classifiers. Advances in Neural Information Processing Systems, 33, 10558-10570.

---

> > ### Comment · Reviewer_g8Bp · 2022-08-08
> > **Thanks for the response.**
> >
> > So, to clarify, on MNIST/CIFAR10, the task you are testing on is a simple classification, and that's what the baseline "Consistency" number is for, right? And the additional "attributes" is something constructed solely for the sensing-reasoning pipeline? How are you training the "sensing" part of the model to predict these additional attributes?
> >
> > I've read the other reviewers' comments and I agree that in the initial revision I read, the overall story could be a bit more coherent. I sympathize with the authors in that they're aiming to do many things, including: 1. establish a new category of models to analyze certified robustness for, 2. establish the difficulty of the problem/a reasonable approach, and 3. demonstrate that the robustness of these models is better than existing approaches. I have not thoroughly read the updated paper yet.
> >
> > Nevertheless, I'll stick with my rating of 6 for now.

---

> > > ### Author Response · Authors · 2022-08-08
> > > **Follow-up Discussion**
> > >
> > > Thanks for the follow-up questions and suggestions!
> > >
> > > > on MNIST/CIFAR10, the task you are testing on is a simple classification, and that's what the baseline "Consistency" number is for, right?
> > >
> > > On MNIST/CIFAR10, we have followed the standard settings for baselines. In particular,  we used LeNet for MNIST and ResNet-110 for CIFAR-10 following the Consistency paper [1]. We report the best certified robustness under different sigma (0.25, 0.50, 1.00) for all methods. We note that the certification results of Consistency in our comparison match the reported results in the original paper Table 1 (CIFAR10) and Appendix Table 2 (MNIST).
> > >
> > > That is to say, here we compare the proposed sensing-reasoning pipeline with the SOTA baseline Consistency [1], and we show that the sensing-reasoning outperforms Consistency method under different radii.
> > >
> > > > the additional "attributes" is something constructed solely for the sensing-reasoning pipeline? How are you training the "sensing" part of the model to predict these additional attributes?
> > >
> > > Thanks for the question. Yes, we construct the “attributes” as knowledge rules for sensing-reasoning pipeline, since one key of the sensing-reasoning pipeline is that we are able to integrate additional *knowledge* into the learning process via the reasoning component. In some tasks, such as PrimateNet and Road sign classification, we can directly use some natural “domain knowledge”, but for MNIST/CIFAR, there lacks direct domain knowledge so we construct some knowledge rules to build the sensing-reasoning pipeline to demonstrate its effectiveness.
> > >
> > > For the sensing part, we train a sensor for each additional attribute. For instance, if we construct attribute A for classes {0, 1, 4, 5}), we will train a sensing model to predict whether a given input belongs to classes {0, 1, 4, 5}. Similar for other attribute sensors. We will add the detailed settings in our revision to make it more clear, and thanks for the valuable suggestions.
> > >
> > >
> > > [1] Jeong, J., & Shin, J. (2020). Consistency regularization for certified robustness of smoothed classifiers. Advances in Neural Information Processing Systems, 33, 10558-10570.
> > >
> > >
> > > > I've read the other reviewers' comments and I agree that in the initial revision I read, the overall story could be a bit more coherent… I have not thoroughly read the updated paper yet.
> > >
> > > We really appreciate the reviewer reading others’ comments and discussions. Indeed, we have updated our revision to tell a more coherent story, and we have also added a roadmap in the revision to make the logic more clear. In particular, following the suggestions, we have added a roadmap at the end of the introduction (line 93-99) in our revision, and we have emphasized the relationship between our hardness analysis and the proposed certification algorithm (line 175-178, line 222-227). Reviewer ZvBd who has pointed out this question has expressed his satisfaction with our response and revision.
> > > We sincerely hope the reviewer could consider our response and revision when evaluating this work, and thank you very much for your time and valuable comments and suggestions again!

---

> ### Author Response · Authors · 2022-08-02
> **Response to Reviewer g8Bp (2/2)**
>
> >how is the model without the "reasoning" actually arriving at its final prediction?
>
> Thanks for the comment and sorry for the confusion. For tasks such as MNIST, CIFAR, Road Sign classification, and Word50, we train an end-to-end model as the sensing-only model for comparison (the model is trained with SOTA robust training methods). For tasks such as PrimateNet, since in the sensing-reasoning pipeline, there are a bunch of “leaf” sensing models, we directly calculate their average certified robustness for comparison, which makes it a strong baseline as these binary models are usually more robust than a multiclass one (more evaluation details in Appendix F./Evaluation metrics). We have made this clear in our revision and thanks for the comment!

---

### Author Response · Authors · 2022-08-02
**Revision Summary according to the Feedbacks from Reviewers**

We thank all the reviewers for recognizing our work as interesting, theoretically solid and well formulated, with significant experiments on different applications. We thank the reviewers for their valuable suggestions and comments. We have revised our paper based on these thoughtful comments. Specifically, we have made the following major updates and highlighted the changes in blue in our revision:

1. We conducted additional experiments and added new results in Appendix H showing that our sensing-reasoning pipeline achieves the new state-of-the-art certified robustness on standard benchmarks (MNIST and CIFAR10), following Reviewer g8Bp’s suggestion.
2. We added the extension analysis of using Bayesian network (BN) as the reasoning component, including the hardness results of certifying the robustness of BN and the concrete certification algorithm for BN in Appendix J, following Reviewer 7FcD’s suggestion.
3. We added the standard definition of “certified robustness” in Section 3 (Line 171-174) with reference, and provide the formal definition of certified robustness given a sensing-reasoning pipeline in Definition 3, following Reviewer ZvBd’s suggestion.
4. We have added transition sentences in section 3 and 4 to make the paper structure more clear.

All updates are highlighted in blue in our revision. If the manuscript is accepted, the new contents in the current Appendix will be merged into the main text given the extra page limit for the camera-ready version.

We look forward to the discussions with all the reviewers for further improvement of our paper. We really appreciate all reviewers’ feedback!

---

### Meta-Review · Area_Chair_Gbxi · 2022-08-26

**Recommendation:** Accept
**Confidence:** Less certain

**Metareview:**

I agree with zVbd that the paper has too many points. A paper should try to make a single point and then one can write more papers :-). The #P hardness result seems the least interesting of the points. I am also not very interested in the certification of robustness. The real question is can Carlini defeat it. But this paper raises an issue that I have always suspected is important --- are structured labels inherently more robust?  I would certainly want to see what Carlini can do in generating adversarial examples for highly structured labels.  As I believe that robustness for structured labels deserves more attention, and this paper purports to have positive empirical results in that direction with reasonable review scores I will recommend acceptance.


**Award:**

No

---

### Decision · Program_Chairs · 2022-09-14

Accept